# Genetic Profiling of a Cohort of Italian Patients with ACTH-Secreting Pituitary Tumors and Characterization of a Novel *USP8* Gene Variant

**DOI:** 10.3390/cancers13164022

**Published:** 2021-08-10

**Authors:** Donatella Treppiedi, Anna Maria Barbieri, Genesio Di Muro, Giusy Marra, Federica Mangili, Rosa Catalano, Emanuela Esposito, Emanuele Ferrante, Andreea Liliana Serban, Marco Locatelli, Andrea Gerardo Lania, Anna Spada, Maura Arosio, Erika Peverelli, Giovanna Mantovani

**Affiliations:** 1Department of Clinical Sciences and Community Health, University of Milan, 20122 Milan, Italy; donatella.treppiedi@unimi.it (D.T.); anna.barbieri@unimi.it (A.M.B.); genesio.dimuro@unimi.it (G.D.M.); giusy.marra@unimi.it (G.M.); federica.mangili@unimi.it (F.M.); rosa.catalano@unimi.it (R.C.); emanuela.esposito1@unimi.it (E.E.); anna.spada@unimi.it (A.S.); maura.arosio@unimi.it (M.A.); giovanna.mantovani@unimi.it (G.M.); 2Endocrinology Unit, Fondazione IRCCS Ca’ Granda Ospedale Maggiore Policlinico, 20122 Milan, Italy; leleferrante@gmail.com (E.F.); andreea.l.serban@gmail.com (A.L.S.); 3Department of Pathophysiology and Transplantation, University of Milan, 20122 Milan, Italy; marco.locatelli@unimi.it; 4Neurosurgery Unit, Fondazione IRCCS Ca’ Granda Ospedale Maggiore Policlinico, 20122 Milan, Italy; 5Endocrinology, Diabetology and Medical Andrology Unit, Humanitas Clinical and Research Center, IRCCS, 20089 Rozzano, Italy; andrea.lania@hunimed.eu; 6Department of Biomedical Sciences, Humanitas University, 20089 Rozzano, Italy

**Keywords:** Cushing’s disease, ACTH-secreting pituitary tumor, USP8, USP48, mutation

## Abstract

**Simple Summary:**

Cushing’s Disease (CD) is a rare but severe endocrine disorder due to an adrenocorticotropic hormone (ACTH)-secreting pituitary tumor, and pathogenetics remained a puzzling issue for a long time. The recent identification of somatic mutations in the 14-3-3 protein binding motif of ubiquitin specific peptidase 8 gene (*USP8*), present in a consistent subgroup of ACTH-secreting pituitary tumors, have represented a major advance in the understanding of CD pathogenesis. In our cohort of 60 patients we found an incidence of 11.7% of *USP8* recurrent somatic mutations whereas a novel *USP8* variant (G664R) located upstream the canonical *USP8* mutational hotspot was identified in one case. This alteration has never been reported by previous records. The present study provides *USP8* G664R variant in vitro functional characterization in AtT-20 cells and demonstrates its possible implication in ACTH-secreting tumor pathogenesis, contributing to enlarge the genetic landscape of CD.

**Abstract:**

Cushing’s Disease (CD) is a rare condition characterized by an overproduction of ACTH by an ACTH-secreting pituitary tumor, resulting in an excess of cortisol release by the adrenal glands. Somatic mutations in the deubiquitinases *USP8* and *USP48*, and in *BRAF* genes, have been reported in a subset of patients affected by CD. The aim of this study was to characterize the genetic profile of a cohort of 60 patients with ACTH-secreting tumors, searching for somatic mutations in *USP8*, *USP48*, and *BRAF* hotspot regions. Seven patients were found to carry *USP8* somatic mutations in the well-characterized 14-3-3 protein binding motif (*n* = 5 P720R, *n* = 1 P720Q, *n* = 1 S718del); 2 patients were mutated in *USP48* (M415I); no mutation was identified in *BRAF*. In addition, a novel *USP8* variant, G664R, located in exon 14, upstream of the 14-3-3 protein binding motif, was identified in 1 patient. Functional characterization of *USP8* G664R variant was performed in murine corticotroph tumor AtT-20 cells. Transient transfection with the *USP8* G664R variant resulted in a significant increase of ACTH release and cell proliferation (+114.5 ± 53.6% and +28.3 ± 2.6% vs. empty vector transfected cells, *p* < 0.05, respectively). Notably, USP8 proteolytic cleavage was enhanced in AtT-20 cells transfected with G664R USP8 (1.86 ± 0.58–fold increase of N-terminal USP8 fragment, vs. WT USP8, *p* < 0.05). Surprisingly, in situ Proximity Ligation Assay (PLA) experiments showed a significant reduction of PLA positive spots, indicating USP8/14-3-3 proteins colocalization, in G664R USP8 transfected cells with respect to WT USP8 transfected cells (−47.9 ± 6.6%, vs. WT USP8, *p* < 0.001). No significant difference in terms of ACTH secretion, cell proliferation and USP8 proteolytic cleavage, and 14-3-3 proteins interaction was observed between G664R USP8 and S718del USP8 transfected cells. Immunofluorescence experiments showed that, contrary to S718del USP8 but similarly to WT USP8 and other USP8 mutants, G664R USP8 displays an exclusive cytoplasmic localization. In conclusion, somatic mutations were found in *USP8* (13.3% vs. 36.5% incidence of all published mutations) and *USP48* (3.3% vs. 13.3% incidence) hotspot regions. A novel *USP8* variant was identified in a CD patient, and in vitro functional studies in AtT-20 cells suggested that this somatic variant might be clinically relevant in ACTH-secreting tumor pathogenesis, expanding the characterization of USP8 functional domains.

## 1. Introduction

Adrenocorticotropic hormone (ACTH)-secreting pituitary tumors are responsible for a rare condition named Cushing’s disease (CD). In affected patients, ACTH oversecretion leads to an excess of cortisol production in the adrenal glands; chronic hypercortisolism results in many severe clinical manifestations such as cardiovascular disease, diabetes, hypertension, osteoporosis and fractures, infections, and thromboembolic events [1]. The prevalence of CD is around 30–40 cases per million inhabitants per year and the incidence is 1–2 per million per year; the female:male ratio is 3:1 and the age of onset mainly occurs during the fourth to sixth decades of life. [1] The first-line treatment for CD is transsphenoidal surgery excision of ACTH-secreting tumor from the pituitary. After surgery, remission is observed in about 70–90% of patients, with a 15–25% of recurrence risk and a 20–30% persistence risk [1].

The pathogenetics of CD remained a puzzling issue for a long time, principally because the search for responsible candidate genes was hampered by the small size of surgical tumors and by the rareness of the disease. An important breakthrough arrived thanks to the recent identification of mutated genes in ACTH-secreting tumors by exome sequencing. In 2015, two independent groups described gain-of-function mutations in the ubiquitin specific peptidase 8 (*USP8*) gene [2,3]. Ubiquitination is a posttranslational reversible process which consists of the binding of ubiquitin (Ub) molecules to specific proteins; once ubiquitinated, peptides are addressed to proteolytic degradation by the 26S proteasome complex [4]. Deubiquitinating enzymes (DUBs) can remove conjugated Ub molecules from target proteins, thus preventing their degradation. USP8 deubiquitinating activity is modulated through the binding of 14-3-3 proteins to the consensus 14-3-3 binding motif RSYSSP at amino acid positions 715-720 [5,6,7,8]. This binding prevents the proteolytic cleavage of USP8 at position 715, which releases the C-terminal 40-kDa deubiquitinase catalytic domain (USP8-C40). Interestingly, the majority of the mutations identified in *USP8* gene are located in the 14-3-3 binding site and impede the binding between USP8 and 14-3-3 proteins; as a consequence, USP8 mutant proteins are more prone to undergo proteolytic cleavage and an abnormal amount of USP8-C40 is produced [2,3]. It has been hypothesized that the increased deubiquitinating activity alters the degradation process of endocytosed EGFR, recycling elevated amount of EGFR on the plasma membrane; *POMC* gene transcription is consequently strengthened by high EGFR levels, leading in turn to an increase in ACTH secretion and to corticotroph tumorigenesis. Subsequently, in 2018, recurrent somatic mutations were identified in *USP48* and *BRAF* genes [9]. So far, *USP48* mutations concern M415, a residue localized in a very well conserved position in the catalytic domain of deubiquitinases; in vitro studies showed that M415I or M415V mutants have a higher deubiquitinating activity, stimulate *POMC* transcription, and potentiate CRH-induced ACTH synthesis. [9,10]. As regards *BRAF*, it harbors the well-known V600E mutation. It has been observed that V600E variant has an increased kinase activity, resulting in activation of MAPK pathway and elevation of *POMC* transcription and ACTH synthesis [9].

Following the identification of driver mutations in ACTH-secreting tumors, in the past few years many groups reported the results of genetic analysis of CD patients, mainly performed for *USP8* by exome sequencing or targeted sequencing [11,12,13,14,15,16,17,18,19,20,21,22,23,24]. The aim of the present study was to characterize the genetic profile of a cohort of 60 patients subjected to surgery excision of ACTH-secreting tumors, searching for somatic mutations in *USP8, USP48*, and *BRAF* hotspot regions. In addition to the classical mutations in deubiquitinases *USP8* and *USP48*, here we describe a novel *USP8* variant located in exon 14, upstream of the 14-3-3 protein binding motif and its functional characterization, contributing to enlarging the genetic landscape of CD.

## 2. Materials and Methods

### 2.1. Sample Collection

ACTH-secreting pituitary tumor samples from 60 subjects were collected during transsphenoidal surgery (TSS) between 1999 and 2020 in our center, and all patients gave written informed consent to the use of their sample. DNA was isolated from frozen specimens (*n* = 54) or from formalin fixed specimens (*n* = 16); 10 samples out of 60 were available both frozen and formalin fixed. 3 patients with persistence of disease underwent a second TSS and samples from both interventions were analyzed.

### 2.2. DNA Extraction and Sanger Sequencing

DNA from frozen specimens was obtained using the Gentra Puregene Tissue Kit (Qiagen, Hilden, Germany); DNA from FFPE samples was isolated by the QIAamp DNA FFPE Tissue Kit (Qiagen, Hilden, Germany). DNA quality and concentration were assessed using a NanoDrop 2000 spectrophotometer (Thermo Fisher Scientific, Waltham, MA, USA). For *USP8* hotspot (exon 14), *USP48* hotspot (exon 10), and *BRAF* hotspot (exon 15) amplifications, couples of primers were designed both in the introns (in order to include the whole exon and the intron-exon splice junctions) and in the exons (PCR primers sequences are listed in Appendix A). COLD-PCR was carried out in a C1000 Thermal Cycler (Bio-Rad, Hercules, CA, USA): after an initial denaturation at 94 °C, 10 cycles were performed with an annealing temperature of 57 °C, followed by 25 cycles with a reduced annealing temperature of 55 °C [25]. PCR products were sequenced using the BigDye Terminator v3.1 Cycle Sequencing Kit and the 3130xl Genetic Analyzer (Applied Biosystems, Foster City, CA, USA).

### 2.3. Plasmids and Mutagenesis

The pME-Flag plasmid containing the full-length wild-type cDNA of human *USP8* with the N-terminal of the *USP8* sequence fused to the Flag tag was kindly gifted by M. Theodoropoulou. The QuickChange II Site-Directed Mutagenesis Kit (Agilent Technologies, Santa Clara, CA, USA) was used to obtain the G664R USP8 and the S718del USP8 plasmids, according to the manufacturer’s instructions. For the USP8-C40 construct, a fragment corresponding to amino acids 715-1118 of human wild-type *USP8* was cloned, starting from human total osteoblast RNA. Briefly, RNA was extracted by RNeasy Plus Mini Kit (Qiagen, Hilden, Germany), and reverse transcribed by RevertAid H Minus First Strand cDNA Synthesis Kit (Thermo Fisher Scientific, Waltham, MA, USA). RT-PCR was carried out with Platinum SuperFi DNA Polymerase, according to the manufacturer’s guidelines (Thermo Fisher Scientific), in a C1000 Thermal Cycler (Bio-Rad, Hercules, CA, USA). Direct sequencing of all constructs was performed using the BigDye Terminator v3.1 Cycle Sequencing Kit and the 31030xl Genetic Analyzer (Applied Biosystems, Foster City, CA, USA). All the primers used are available on request.

### 2.4. ACTH-Secreting Pituitary Cell Culture

Murine pituitary corticotroph tumor cells, AtT-20 cells (ATCC CRL-1795™), were cultured in DMEM (Life Technologies, Carlsbad, CA, USA) supplemented with 10% fetal bovine serum (FBS) and antibiotics (Life Technologies, Carlsbad, CA, USA). Cells were cultured at 37 °C in a humidified atmosphere with 5% CO_2_.

### 2.5. Cell Transfection

AtT-20 cells were seeded in 6-well plates at a density of 4 × 10^6^/well. The day after, cells were transiently transfected with expression vectors or empty vector for indicated times with Lipofectamine 2000 reagent (Invitrogen, Thermo Fisher Scientific, Waltham, MA, USA), according to the instructions of the manufacturer. For each experiment, transfection efficiency was monitored by Western blot analysis using an anti-FLAG antibody (Merck, KGaA, Darmstadt, Germany). Empty vector was used in each experiment as negative control (mock). Original western blot images are available in Appendix A.

### 2.6. Western Blot Analysis

For phospho-CREB, p27/kip1, and USP8 proteolitic cleavage analysis, AtT-20 cells were transiently transfected for 96 h. For phopsho-ERK1/2 evaluation, AtT-20 cells were transiently transfected with USP8 constructs for 48 h and treated with 50 ng/mL EGF for 48 h in medium supplemented with 2% FBS. Cells were lysed with lysis buffer and total proteins were quantified by BCA assay. 30–100 µg of total proteins extracted were separated on SDS/polyacrylamide gels and transferred to a nitrocellulose filter. p27 antibody was from Santa Cruz Biotechnology (Dallas, TX, USA) and diluted at 1:200. Antibodies against phospho-CREB and total CREB, phospho-ERK1/2, and total ERK1/2 were from Cell Signaling Technology (Danvers, MA, USA) and used at 1:1000 dilution. The anti-Flag antibody (Merck KGaA, Darmstadt, Germany) was used at 1:1000 dilution. The USP8 antibody (against a synthetic peptide equivalent to the N-terminal of USP8) was from Merck KGaA (Darmstadt, Germany) and used at 1:1000 dilution to recognize both the full-length form of USP8 and the N-term 90 kDa USP8 fragment. Primary antibodies were incubated o/n at 4 °C, secondary antibodies anti-rabbit or anti-mouse (Cell Signaling Technology, Danvers, MA, USA) were used at 1:2000 and incubated at room temperature for 1 h. For normalization, anti-GAPDH antibody (Ambion, Thermo Fisher Scientific, Waltham, MA, USA) was used at 1:4000 for 1 h at room temperature. Chemiluminescence was detected using a ChemiDOC-IT Imaging System (UVP, Upland, CA, USA) and densitometrical analysis was performed with NIH ImageJ software (National Institutes of Health, Bethesda, MD, USA).

### 2.7. Hormone Levels Detection

At 72 h of transfection, AtT-20 cells were trypsinized, counted, and re-seeded in a 24-well plate at a density of 12.5 × 10^4^ cells/well in 500 µL of culture medium at 37 °C. The following day culture medium was replaced with 300 µL of fresh medium, and cells were incubated at 37 °C for 4 h based on preliminary time course experiments. Culture media were then collected to detect secreted ACTH. Murine ACTH levels were determined using a specific Elisa immunoassay kit (Fine Test, Wuhan Fine Biotech Co., Ltd., Wuhan, China), according to the manufacturer’s instructions and previously reported protocols [26]. Absorbance was read at 450 nm in a Victor2 multilabel plate reader (Perkin Elmer, Whaltam, MA, USA). Data were plotted and analyzed with Curve Expert 1.4 program. Hormone detection was done in triplicate for each condition and experiments were replicated three times. Hormone levels were normalized on the protein content, measured by BCA assay. To test transfection efficiency, Western blot analysis was carried out before hormone determination, with an anti-FLAG antibody for each experiment.

### 2.8. Immunofluorescence Analysis and In Situ Proximity Ligation Assay

AtT-20 cells were seeded on 13-mm poly-L-lysine coated coverslips at a density of 1.25 × 10^5^ cells/well in 24-well plates and grown at 37 °C for 18 h. The following day cells were transiently transfected for 6 h, then chilled on ice and fixed with 4% paraformaldehyde (Sigma-Aldrich, St. Louis, MO, USA) for 10 min at room temperature, washed three times with PBS, and incubated for 1 h at room temperature with blocking buffer (5% FBS, 0.3% Triton™X-100, in PBS). Cells were incubated overnight at 4 °C with anti-FLAG (1:250, Merck KGaA, Darmstadt, Germany) and anti-14-3-3 (1:200, Cell Signaling Technology, Danvers, MA, USA) antibodies. For immunofluorescence analysis, anti-mouse Alexa Fluor™-546-conjugated secondary antibody (1:500, ThermoFisher Scientific, CA, USA) and anti-rabbit Alexa Fluor™-488-conjugated secondary antibody (1:500, ThermoFisher Scientific, CA) were incubated at room temperature for 1 h. All antibodies were diluted in Antibody Diluent Reagent Solution (Life Techologies, ThermoFisher, CA, USA). In order to detect potential unspecific signal, negative control coverslips were incubated with only one primary antibody, respectively.

For in situ Proximity Ligation Assay (PLA) a commercially available kit was used (Duolink; Sigma-Aldrich, St. Louis, MO, USA), accordingly to previous protocols [27]. For both applications, coverslips were mounted on glass slides with Duolink In Situ Mounting Medium with 4’,6-diamidino-2-phenylindole (DAPI) (Sigma-Aldrich, St. Louis, MO, USA) for subsequent observation at fluorescence microscope. Proximity ligation events were quantified with NIH ImageJ software [27]. Briefly, cells were first searched using an excitation light of 405 nm (DAPI signal) and fine focus adjustment was performed switching the filter of the excitation light to FITC (488 nm). This procedure minimized photobleaching before image acquisition of PLA signals. Images of cells showing PLA spots were subjected to deconvolution protocols before quantification in order to exclude any possible unspecific signal from the analysis. Ten fields for each transfection condition were randomly chosen from three independent experiments. About 150 cells showing PLA signals were analyzed for each condition for each experiment. An approximate total of 1215 ± 103 PLA spots were quantified for USP8 WT, with a mean of 7.2 ± 0.5 PLA events/cell, which was then considered as a reference for S718del USP8 and G664R USP8.

### 2.9. Cell Proliferation Assay

Cell proliferation was assessed by colorimetric measurement of 5-bromo-2-deoxyuridine (BrdU) incorporation during DNA synthesis in proliferating cells. 72 h transiently transfected AtT-20 cells were trypsinized, counted, and re-seeded in starved medium in 96-well plate at a cell density of 5 × 10^5^ cells/well. 24 h later, BrdU incorporation was allowed for 2 h. Cell proliferation assay was conducted following the instruction of the manufacturer (GE Healthcare, Life Science, Buckinghamshire, UK), each determination was done in triplicate, and experiments were replicated three times.

### 2.10. Statistical Analysis

The results obtained with AtT-20 cells are expressed as the mean ± S.D. A paired two-tailed Student’s t-test was used to detect the significance between two series of data. *p* < 0.05 was accepted as statistically significant. Patients’ data are expressed as mean ± S.D. for normally distributed continuous variables, median and interquartile range (IQR), or range for non-Gaussian data and proportion for categorical parameters. The latter were analyzed using the χ^2^ test or the Fisher exact test if the expected value was < 5. Continuous parameters with normal distribution were compared using the t test and non-Gaussian data using the non-parametric test of Mann Whitney. Two-sided *p*-value was considered statistically significant when < 0.05. All statistical analyses were performed using SPSS, version 26 (IBM, Chicago, IL, USA).

## 3. Results

### 3.1. Clinical Characteristics and Prevalence of USP8 and USP48 Mutations in Patients

We performed a retrospective study on a cohort of 60 patients diagnosed with CD who underwent TSS between 1999 and 2020 in our Centre. Forty-five out of sixty patients have been described in a recent paper which analyzed the surgical outcome of subjects with CD after TSS between 1990 and 2016 [28]. Clinical characteristics and hormonal status of patients are summarized in Table 1. The female:male ratio was 3:1 (45 females and 15 males); the mean age at diagnosis was 45 ± 15 years. MRI analysis showed 9 patients (15%) with macroadenomas, 39 patients (65%) with microadenomas, and 12 subjects (20%) with negative imaging. The remission rate after surgery was 77.9%. Due to persistence of the disease, 3 patients underwent a second neurosurgery procedure, and one subject underwent a third.

To assess the presence of somatic mutations in *USP8* gene, we decided to sequence the whole exon 14 and the corresponding exon-intron junctions, since currently 26 different mutations have been described in the hotspot region and nearby. Seven mutations were identified in the 14-3-3 protein binding motif: p.Pro720Arg (five patients), p.Pro720Gln (one patient), and p.Ser718del (one patient). The mutation frequency was 11.7%. As regards *USP48* gene, we targeted sequenced the region encompassing Met415, the only amino acidic residue affected by mutations in patients, founding the mutation pMet415Ile in two patients. The mutation frequency was 3.3%. Regarding *BRAF* gene, we did not find the classical pathogenic variant p.Val600Gln in any subject of our cohort. All the mutations identified were heterozygous and were not present in the germline. In any of the tumors analyzed, mutations in *USP8* and *USP48* were present contemporarily.

Patients affected by *USP8* mutation had a higher frequency of macroadenoma (4/7 vs. 4/46, *p* = 0.005) than WT patients, and a higher CD recurrence rate during follow-up (50% vs. 13.1%, *p* = 0.009). Age, sex, ACTH, and UFC levels at baseline, as well as remission rate after surgery, did not differ in the two groups (Table 1).

The *USP48* carriers were two females aged 31 and 25 years at CD diagnosis. The first one had a macroadenoma, successfully removed with surgical treatment; however, the patient experienced disease recurrence six months later. The second patient had a microadenoma on MRI and manifested persistent hypercortisolism after pituitary surgery.

### 3.2. Identification of a New USP8 Variant

Searching for somatic mutations in our cohort of CD patients, in one subject we found a G to A transition at nucleotide position 2165 (c.2165G>A; GenBank accession number NM_005154.5), which results in the substitution of arginine for glycine at amino acid position 664 (p.Gly664Arg) (Appendix A).This novel variant is located in exon 14, upstream from the 14-3-3 protein binding motif (Figure 1a). Homology alignment of the *USP8* region surrounding the p.Gly664 revealed that this zone is highly conserved across a variety of vertebrate species, suggesting the importance and a possible role for the involved amino acids (Figure 1b). So far, structural studies on USP8 protein have focused on specific domains, such as the catalytic domain or the rhodanese domain, for which the crystal structures are available. More recently the interaction between a peptide corresponding to the USP8/14-3-3 binding motif (from p.Lys712 to p.Glu724) and the 14-3-3 protein has been elucidated by crystallography. At present, a crystal model of the region encompassing the p.Gly664 does not exist. In order to identify proteins containing a similar amino acidic region and for which a 3D structural model was available, we performed a protein BLAST search using as query the USP8 region surrounding the p.Gly664; the search did not evidence any homologous protein domain. Thereafter, to understand whether the p.Gly664Arg variant had pathogenic effects on the structure and function of USP8, we did an in silico analysis with some prediction software. In PolyPhen-2 (Harvard Medical School, Boston, MA, USA), the missense p.Gly664Arg variant has a score of 0.997 (sensitivity: 0.0; specificity: 1.0), where scores greater than 0.908 are predicted to be probably damaging; to make a comparison, the p.Pro720Arg mutation has a score of 0.994. Using MutationAssessor (Computational Biology Center, Memorial Sloan Kettering Cancer Center, New York, NY, USA), a software which predicts deleterious effects caused by a mutation evaluating the evolutionary conservation of the affected amino acid in protein homologs, the p.Gly664Arg variant has a score of 0.711 and the p.Pro720Arg mutation has a score of 0.561 (score between 0 and 1, with higher scores more likely deleterious). As regards population genetics, the p.Gly664Arg variant was never identified in any individual out of 400.000.

### 3.3. Overexpression of USP8 G664R Variant in AtT-20 Cells Increased ACTH Secretion and Cell Proliferation

In vitro functional studies of the new *USP8* G664R variant were performed in mouse corticotroph tumor cells AtT-20, bearing wild-type *USP8* and endogenously expressing EGFR (Appendix A).

As an initial step, we examined the impact of *USP8* G664 variant on ACTH release. To this purpose, AtT-20 cells were transiently transfected with WT, S718del, G664R USP8, and USP8-C40. S718del was chosen as the most representative USP8 mutant since it has shown the higher in vitro ability to induce *POMC* transcription and ACTH production when transfected in AtT-20 cells compared to other USP8 mutants [2,10]. USP8-C40 was used as a catalytically active mutant. We found that cells expressing G664R USP8 had increased ACTH secretion compared to empty vector (mock) control cells (+114.5 ± 53.6, *p* < 0.05) and those expressing WT USP8, and had similar ACTH secretion compared to those expressing S718del USP8 and USP8-C40 (Figure 2a). No difference in ACTH secretion was observed in WT USP8 and empty vector transfected cells.

To determine possible implications of USP8 G664R variant on cell proliferation, we performed BrdU incorporation assays. Our results showed that the transfection of USP8 G664R in AtT-20 cells promoted cell proliferation (+28.3 ± 2.6%, *p* < 0.001 vs. mock), similarly to S718del USP8 (+42.8 ± 22.4%, *p* < 0.05 vs. mock) and USP8-C40 (+24.6 ± 16.2%, *p* < 0.05 vs. mock) (Figure 2b).

Since USP8 mutants pathogenetic mechanism involves an abnormal activation of ERK due to a persistent activation of EGFR, we evaluated the level of active phosphorylated ERK1/2 in cells transfected with WT and mutants USP8 upon EGF stimulation. We found that the EGF ability to stimulate ERK phosphorylation in WT USP8 transfected cells was transient with a peak of phosphorylation reached at 24 h and strongly reduced at 48 h incubation (data not shown). In contrast, in cells expressing S718del and G664R USP8, a persistent activation of ERK was observed at 48 h incubation with EGF (+71.5 ± 10.5% S718del USP8 vs. mock, *p* < 0.05, +67.6 ± 34.7%, G664R USP8 vs. mock, *p* < 0.05) (Figure 2c).

The transcription factor CREB is known to be involved in *POMC* promoter activation [29] and ACTH release from pituitary corticotroph cells [30], and its phosphorylated form has been recently found increased in *USP8* mutated tumors and AtT-20 cells transfected with USP8 P720R mutant [19]. Thus, we tested whether USP8 G664R variant could modulate the level of phospho-CREB in AtT-20 cells. We observed a significant increase of CREB phosphorylation in cells transfected with both G664R and S718del USP8 (+28.8 ± 5.5%, G664R USP8 vs. mock, *p* < 0.01; +39.3 ± 22%, S718del USP8 vs. mock, *p* < 0.05), but not in WT USP8 cells (Figure 2d).

As final step, we assessed the expression level of p27/kip1, a key tumor suppressor gene involved in cell cycle regulation in the corticotrophs [31,32]. Particularly, downregulation of p27/kip1 has been recently documented in corticotroph tumors carrying the *USP8* mutations [19]. Western blot analysis performed in AtT-20 cells transiently transfected with G664R, S718del, WT USP8, and empty vector revealed that p27/kip1 expression was strongly reduced in cells transfected with both G664R and S718del USP8 (−30.7 ± 14.9%, *p* < 0.01 and −8 ± 22.7%, *p* < 0.05, respectively) but not in WT USP8 cells (Figure 2e).

### 3.4. USP8 G664R Variant Leads to USP8 Proteolytic Cleavage by Affecting USP8/14-3-3 Proteins Binding

Since downstream effects induced by the new *USP8* G664R variant were similar to those induced by classical *USP8* mutation S178del, we investigated the molecular mechanisms involved. Indeed, G664 is not localized in the 14-3-3 binding site where all the known *USP8* mutations are found.

The increased catalytic activity of the USP8 mutants so far described is due to an USP8 increased proteolytic cleavage into an N-terminal 90-kDa and C-terminal 40-kDa fragments. Therefore, we evaluated whether *USP8* G664R variant could also affect USP8 cleavage. To this aim, we performed immunoblotting experiments using an USP8 antibody raised against the N-terminal region of USP8, recognizing the 90-kDa fragment. As shown in Figure 3a, G664R variant conferred to USP8 a higher susceptibility to cleavage with respect to WT USP8 (+2.01 ± 0.48 fold, *p* < 0.05 vs. WT USP8), similarly to what was observed with the S718del mutant.

The thus-far described *USP8* mutations increased USP8 cleavage by abolishing USP8 binding with 14-3-3 proteins. We thus examined possible alterations of G664R USP8 binding to 14-3-3 proteins by In situ Proximity Ligation Assay (PLA). PLA experiments were carried out on AtT-20 cells endogenously expressing 14-3-3 proteins and transiently transfected with G664R, S718del, and WT USP8. First, by Western blot analysis, we verified that cells transfected with USP8 plasmids showed similar increased expression of USP8 (). As expected, our data demonstrated that USP8 S718del transfected cells have a reduction of PLA positive spots, indicating proteins colocalization, compared with WT USP8 transfected cells (−64.5 ± 5.2%, *p* < 0.01), and, surprisingly, revealed a significant similar decrease in the PLA efficiency in G664R USP8 transfected cells with respect to WT USP8 (−47.9 ± 6.6%, *p* < 0.01) (Figure 3b,c).

We also investigated possible variations in USP8 intracellular localization due to G664R variant. Immunofluorescence experiments performed on transfected AtT-20 cells showed that G664R USP8 is predominantly located within the cytosol, similarly to WT USP8 (Figure 3d). In contrast, S718del USP8 displayed an exclusive nuclear distribution, as previously described in Hela cells [2].

### 3.5. Clinical Significance of Novel USP8 Variant

The *USP8* G664R variant was found in a 21-year old male with an ACTH-secreting microadenoma. In this patient, pituitary surgery allowed to obtain the remission of disease. The results of comparative study between WT and *USP8* mutations did not differ when also considering this subject in the group of mutated patients.

## 4. Discussion

In 2015, the discovery of gain-of-function mutations in the deubiquitinase gene *USP8* represented a major breakthrough in the understanding of pathogenesis of CD [2,3]. All *USP8* mutations identified occur in exon 14 and result in disrupted USP8 association with 14-3-3 proteins, enhanced USP8 proteolytic cleavage, and constitutive DUB activity. Since then, genetic sequencing of the *USP8* mutational hotspot has been carried out on hundreds of corticotropinomas [10,33,34].

Our study population consisted of 60 (45F/15M) patients who were evaluated retrospectively. We identified 3 previously reported *USP8* mutations in 7/60 tumors (p.Pro720Arg, p.Pro720Gln, and p.Ser718del) with an overall prevalence of 11.7%. The frequency of *USP8* mutations observed in our cohort is lower than in most cohorts described so far. Indeed, according to a recent published meta-analysis, *USP8* somatic mutations have been reported in about 35% (20–60%) of cases [10], with the lowest and highest prevalence registered in European studies [16,22,34] and a large Chinese cohort [3], respectively, probably suggesting a role of genetic background and ethnicity differences. Mutations in *USP48* and *BRAF* have also been recently detected in corticotroph tumors, mostly wild type for *USP8* itself [9,24,35]. We found *USP48* mutations in 2/60 patients, whereas none of the subjects were mutated for *BRAF*, still being both percentages lower if compared to other studies [9,24].

From a clinical point of view, more than half of USP8 mutation carriers had an ACTH secreting macroadenoma and were at a higher risk of recurrence compared to WT patients, although our analysis could have been affected by the small number of patients. Previous reports regarding tumor size and aggressive behavior of disease showed variable findings, although overall data did not show a significant difference [18]. Our clinical data are in line with those of Martins and colleagues, showing that USP8 mutated tumors have a superior tumor size but a similar clinical and biochemical phenotype to wild-type tumors [21].

Moreover, the higher recurrence rate could be a consequence of the radiological characteristics of *USP8* mutated patients. In this context, a recent meta-analysis showed that macroadenomas are associated with a higher relapse rate of hypercortisolism after pituitary surgery [36].

While screening for *USP8* mutations, we found a novel *USP8* genetic variant p.Gly664Arg (G664R) in one corticotropinoma. A possible explanation as to why this *USP8* variant was never reported in the literature might rely on the fact that some of the previously published works used a primer encompassing the 664 residue [34,37] or a primer designed downstream of the 664 residue [19,20,22] for sequencing.

The G664R variant is associated with a high PolyPhen-2 score, indicating that this amino acid substitution could have a strong probability in determining a damaging impact on the structure and function of USP8 protein. These evidences prompted us to perform a functional characterization by means of AtT-20 cells, a murine ACTH secreting cell line bearing wild type *USP8*. The effects played by G664R USP8 were compared with those exerted by S718del, one of the USP8 mutants that has shown the highest activity in vitro in terms of *POMC* transcription and ACTH production, and by USP8 catalytically active fragment USP8-C40, chosen as further positive control.

Firstly, we observed an increased ACTH production by cells expressing G664R USP8 compared to those expressing WT USP8, this result being in line with data obtained by Reincke and colleagues in cells transfected with S718del and USP8-C40 [2]. Then, by measuring BrdU incorporation into the newly synthetized DNA of proliferating cells, we showed that G664R USP8 expression in AtT-20 cells promoted cell proliferation similarly to what was observed in cells expressing S718del USP8 or USP8-C40.

USP8 mutations have been shown to be associated with unbalanced EGFR signaling, enhanced phosphorylation of the downstream effector ERK1/2, and increased AP1 transcriptional activity on POMC promoter [2,3]. In agreement with previous results, overexpression of USP8 G664R also resulted in high levels of active phosphorylated ERK1/2 after 48 h of EGF stimulation, thus confirming that the novel USP8 variant sustains EGFR-MAPK signaling to promote corticotrophs ACTH production and cell growth. Indeed, in the tumoral corticotrophs, components of the MAPK pathway have been found to be overactivated and correlated to cell proliferation promotion [38,39,40]. It is interesting to note that, while AtT-20 cells used by Fukuoka et al. and Reincke et al. did not express endogenous EGFR [2,41], our AtT-20 subclone endogenously expressed the receptor. Similarly, Asari and coworkers used an AtT-20 subclone expressing EGFR [42].

However, EGFR is not the solely target of USP8. Recent works have shown deregulation of several molecules in *USP8*-mutated tumors compared to wild type tumors [17,19,21]. Particularly, our results confirmed that p27/Kip1 and phosphorylated CREB expression levels were respectively reduced and increased in cells transfected with USP8 G664R variant, in line with the proliferative and secretive role played by this novel USP8 variant and at a similar extent as S718del USP8. However, whether p27/Kip1 and CREB are USP8 direct or indirect clients remains to be established. It also has to be mentioned that in a Brazilian cohort, no difference in the gene expression of markers involved in cell cycle regulation including p27/Kip1 has been found between wild-type and USP8 mutated groups [21], suggesting that altered USP8 deubiquitinase activity might influence other pathways related to cell growth and proliferation. At the same time, gene expression profiles analysis in *USP8* mutant and wild-type tumors revealed that counter regulatory mechanisms may fine-tune the EGFR-MAPK pathway in tumor corticotroph cells [17]. Testing the hypothesis of corticotrope-specific USP8 substrates other than EGFR has provided remarkable insight into the biological behavior of these cells. Indeed, the recent demonstration of the ACTH biosynthetic pathway as a direct target of ubiquitination has raised the idea of targeting the ubiquitin proteasome system to modulate ACTH turnover both in *USP8* mutant and wild-type tumors [17].

Furthermore, and surprisingly, our Western blot and PLA experiments demonstrated that the USP8 G664R variant shares the same molecular mechanisms underlying its biological effects exerted by other canonical USP8 mutants, such as a diminished binding to 14-3-3 proteins conferring to USP8 a higher susceptibility to proteolytic cleavage. It is noteworthy that the substitution of an arginine for the highly conserved glycine 664 results in the change of a polar residue for a non-polar amino acid. The arginine residue, providing an extra charge or more simply because of its steric hindrance, may alter the conformational structure of USP8, thus preventing the binding to 14-3-3 proteins. Several studies described glycine to arginine disease-associated mutations, confirming the pathological relevance of these amino acidic changes in causing dramatic structural and functional anomalies [43,44,45,46].

To the best of our knowledge, only a few other individual *USP8* variants located completely outside the 14-3-3 binding site (p.Thr735Ile, Asn741Asp) were reported in previous studies [3,22]. However, their functional impact needs further investigation.

*USP8*-mutated pituitary tumors mostly displayed high immunoreactivity of USP8 in the nuclei [2,19]. This observation prompted speculation that nuclear localization of USP8 may indicate the presence of USP8 mutations in corticotropinomas. Our immunofluorescence data in AtT-20 cells indicated instead that USP8 G664R is predominantly located within the cytosol, as the wild-type protein. It must be mentioned that in vitro studies on Hela cells have shown that some of the USP8 mutants, such as S718P and P720R, are localized both in the nucleus and in the cytoplasm [2]. These results suggest that subcellular localization of mutant USP8 is heterogeneous. Unfortunately, due to lack of material, we could not test the immunoreactivity of G664R USP8 in the tumor sample of the patient, thus the intracellular localization of G644R USP8 in human corticotropinomas remains to be fully demonstrated.

## 5. Conclusions

In conclusion, somatic mutations were found in *USP8* (13.3% vs. 36.5% incidence of all published mutations) and in *USP48* (3.3% vs. 13.3% incidence) hotspot regions. In our cohort, mutated tumors seem to be associated with a more aggressive clinical phenotype (i.e., higher prevalence of macroadenoma as well higher recurrence rate of disease). Moreover, the present study provides an in vitro functional characterization of the novel G664R *USP8* genetic variant firstly identified in one patient with CD and located outside the USP8 mutational hotspot and demonstrates its possible implication in ACTH-secreting tumor pathogenesis.

## Figures and Tables

**Figure 1 cancers-13-04022-f001:**
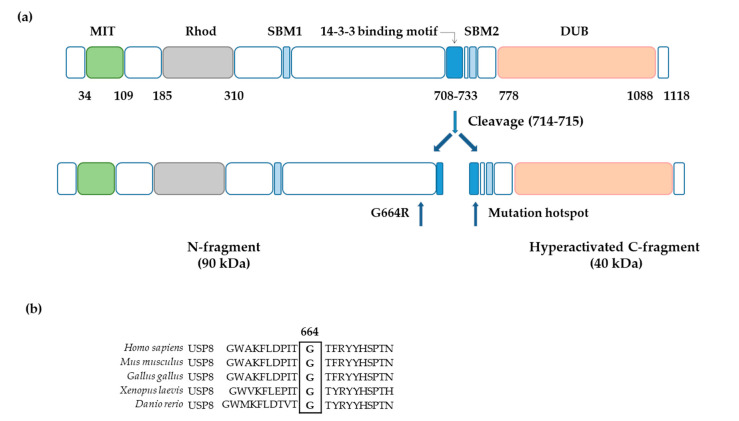
Identification of a novel *USP8* somatic variant in ACTH-secreting pituitary tumors. (**a**) The novel *USP8* G664R variant is located upstream the *USP8* mutational hotspot (amino acid 713-720). Human USP8 functional regions: MIT domain (MIT); rhodanese-like domain (Rhod); SH3-binding motif (SBM); 14-3-3 binding motif (14-3-3); and deubiquitinase catalytic domain (DUB)**.** (**b**) Homology alignment of USP8 amino acidic sequence: the boxed glycine at residue 664 is highly conserved between vertebrate species.

**Figure 2 cancers-13-04022-f002:**
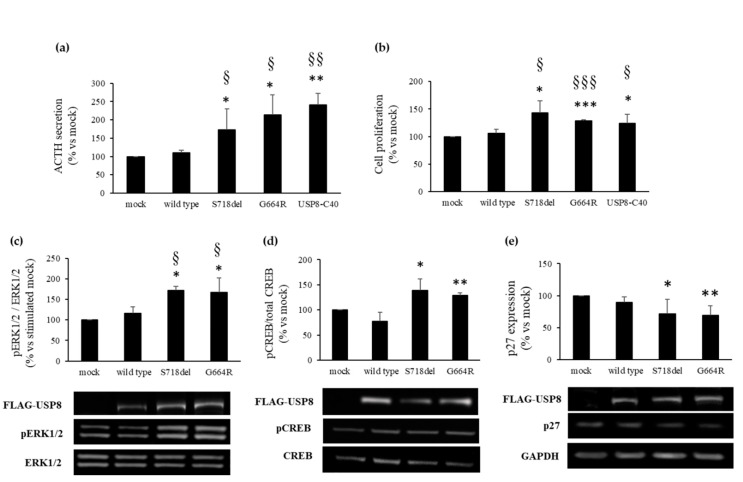
Effects of USP8 G664R variant on hormone secretion and cell proliferation. (**a**) ACTH released in culture medium from AtT-20 cells transiently transfected with USP8 constructs was measured by competitive ELISA kit. Each determination was done in triplicate. (**b**) Cell proliferation results of AtT-20 cells transfected with USP8 constructs by BrdU assay. Each determination was done in quintuple. (**c**) Graphs showing densitometrical analysis of p-ERK1/2 expression levels in AtT-20 cells transiently transfected with USP8 constructs and stimulated with EGF 50 nM for 48 h. (**d**,**e**) Graphs resulting from densitometrical analysis of p-CREB and p27/kip1 expression levels in AtT-20 cells transiently transfected with USP8 constructs. Values represent mean ± SD and results are expressed as percentage of empty vector transfected cells (mock). *, *p* < 0.05, **, *p* < 0.01, ***, *p* < 0.001 vs. mock; §, *p* < 0.05, §§, *p* < 0.01, §§§, *p* < 0.001 vs. WT USP8 transfected cells.

**Figure 3 cancers-13-04022-f003:**
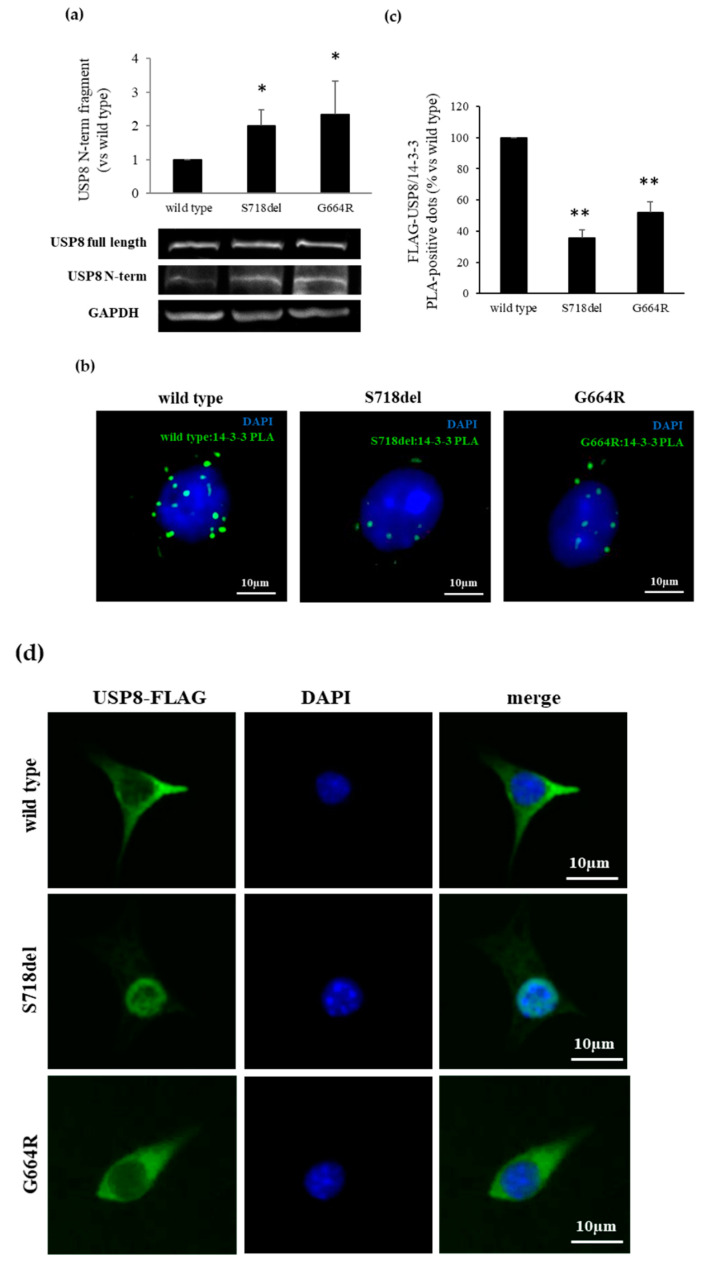
G664R substitution increases proteolytic cleavage of USP8 hampering its interaction with 14-3-3 proteins. (**a**) Representative Western blot experiment and densitometrical analysis showing enhanced USP8 N-term fragment production in AtT-20 cells transfected with G664R USP8 mutant. *, *p* < 0.05 vs. WT USP8 transfected cells (**b**) Representative in situ PLA experiment performed in AtT-20 cells transfected with USP8 mutants. AtT-20 cells were incubated with anti-FLAG and anti-14-3-3 antibodies, respectively. Mouse anti-FLAG and rabbit anti-14.3. probes were used, respectively. Each picture represents a typical cell staining observed in 10 fields randomly chosen. Green dots represent PLA events and indicate close proximity between USP8 and 14-3-3 proteins. Cell nuclei are stained in blue with DAPI. Scale bars 10 μm. (**c**) Quantification of total USP8/14-3-3 puncta representing PLA events as shown in (**b**), showing a decrease in USP8/14-3-3 interaction in G664R USP8 transfected cells (*n* = 3 and the number of PLA puncta per cell was quantified for 150 cells randomly chosen from different fields per condition, **, *p* < 0.01 vs. WT USP8 transfected cells). (**d**) Representative immunofluorescence experiment showing subcellular localization of USP8 mutants (green) in AtT-20 cells. Nuclei are stained with DAPI (blue). G664R USP8 variant is located exclusively in the cytoplasm. Scale bars, 10 µm.

**Table 1 cancers-13-04022-t001:** Demographic, clinical, and neuroradiological characteristics of CD patients.

	All (*n* = 60)	WT (*n* = 50)	USP8 (*n* = 7) *	*p* ^§^
Age (years, mean ± SD)	45 ± 15.1	46 ± 15.1	46.8 ± 14.8	0.88
Females [*n*/tot (%)]	45/60 (75%)	37/50 (74%)	6/7 (85%)	0.67
ACTH pg/mL (median, IQR)	44.3 (33.1–77.6)	43.6 (32–76)	57 (38–110)	0.308
Relative UFC (median, IQR)	2.3 (1.2–3.7)	2.3 (1.2–3.3)	2.1 (0.6–7.1)	0.832
Microadenoma [*n*/tot (%)]	39/60 (65)	35/50 (70)	2/7 (28)	0.041
Macroadenoma [*n*/tot (%)]	9/60 (15%)	4/50 (8)	4/7 (57.1)	0.005
Negative imaging [*n*/tot (%)]	12/60 (20)	11/50 (22)	1/7 (14.2)	1
Remission [*n*/tot (%)]	46/60 (76.6)	38/50 (76)	6/7 (86.5)	1
Recurrence [*n*/tot (%)]	9/46 (19.5%)	5/38 (13.1)	3/6 (50)	0.009
Follow-up [years, median IQR)]	5 (2–8)	5 (2–8)	4 (3–6)	0.614

* The *USP8* variant and the two *USP48* carriers were described separately; ^§^ WT vs. *USP8*.

## Data Availability

Not applicable.

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
