# Peer review of "Genetic Profiling of a Cohort of Italian Patients with ACTH-Secreting Pituitary Tumors and Characterization of a Novel USP8 Gene Variant"

_cancers, 2021, doi:10.3390/cancers13164022_

Round 1

Reviewer 1 Report

This paper adds to the growing number of studies on USP8 variants in ACTH-secreting adenomas. The series, though less consistent compared to other publications, provides description of a novel variant. Findings indicate that this variant, although outside the mutational hotspot, exerts the same alterations.

- Annotated version for the reference sequence should be provided as according to the current  NM_005154.5, nucleotide at c.2188 is A; location of the reported variant G>R is c.2166.

- The novel USP8 variant appears to exert effects similar to known variants, i.e., S718del, on luciferase-POMC transcription, ERK and CREB phosphorylation, p27 expression, protein cleavage. Given the heterogeneity in mutant USP8 subcellular localization (Reincke 2014, Weigand 2019), the importance of cytosolic USP8 G664R should be placed into proper context.

- sequencing of USP8 has been performed both for exon 14 alone (Hayashi 2016, Losa 2018, Weigand 2019) and across the entire coding sequence, comprising intron-exon regions (see both seminal papers by Reincke 2014 and Ma 2014, Perez Rivas 2015), thus the statement at page 11, 425-429, is incorrect given that G664R is in exon 14.

- No study has detected changes in USP8 gene and protein expression in USP8 mutant adenomas {Weigand 2019, Ma 2014, Bujko 2019, Sesta 2020), thus there is so far no rationale to assume upregulation of USP8 in variant adenomas (see page 12, lines 484-485)

- there are several grammar and conceptual errors (mouse adrenocortical tumor cells AtT-20, page 7, line 292)

- the paper could benefit from discussion of additional findings on USP8 pathophysiology in corticotrope adenomas, e.g., Martins 2020, Sesta 2020

- given the small number of specimens carrying USP8 variants, statistics should be performed with non parametric tests and comparisons on 7 patients described with appropriate caution, especially as regards tumor size and recurrence, known to be linked per se.

Author Response

Answer to reviewer

Reviewer #1

This paper adds to the growing number of studies on USP8 variants in ACTH-secreting adenomas. The series, though less consistent compared to other publications, provides description of a novel variant. Findings indicate that this variant, although outside the mutational hotspot, exerts the same alterations.

1- Annotated version for the reference sequence should be provided as according to the current NM_005154.5, nucleotide at c.2188 is A; location of the reported variant G>R is c.2166.

We changed the location of the variant G664R according to the current (GenBank accession number: NM_00515154.5) reference sequence (page 6, lines 280-281). The nucleotide modified in the patient is 2165, not 2166, since the mutated G is the first in the GGA triplet. We added the number of the current reference sequence in the results, nearby the location of the variant.

2- The novel USP8 variant appears to exert effects similar to known variants, i.e., S718del, on luciferase-POMC transcription, ERK and CREB phosphorylation, p27 expression, protein cleavage. Given the heterogeneity in mutant USP8 subcellular localization (Reincke 2014, Weigand 2019), the importance of cytosolic USP8 G664R should be placed into proper context.

Upon reviewer’s suggestion, the discussion section of the manuscript has been revised to give place the importance of to the role of the subcellular localization of USP8 mutants and USP8 G664R in the proper context.  Page 13, lines 509-519 have been modified as follow:

USP8-mutated pituitary tumors mostly displayed high immunoreactivity of USP8 in the nuclei [2, 19]. This observation prompted to speculate that nuclear localization of USP8 may indicate the presence of USP8 mutations in corticotropinomas. Our immunofluorescence data in AtT-20 cells indicated instead that USP8 G664R is predominantly located within the cytosol, as the wild-type protein. It has to be mentioned that in vitro studies on Hela cells have shown that some of USP8 mutants such as S718P, P720R are localized both in the nucleus and in the cytoplasm [2]. These results suggest that subcellular localization of mutant USP8 is heterogeneous. Unfortunately, due to lack of material, we could not test the immunoreactivity of G664R USP8 in the tumor sample of our patient, thus the cytosolic localization of G644R USP8 in human corticotropinomas remains to be fully demonstrated.”

3- sequencing of USP8 has been performed both for exon 14 alone (Hayashi 2016, Losa 2018, Weigand 2019) and across the entire coding sequence, comprising intron-exon regions (see both seminal papers by Reincke 2014 and Ma 2014, Perez Rivas 2015), thus the statement at page 11, 425-429, is incorrect given that G664R is in exon 14.

We fully agree with the reviewer that Reincke (2015), Ma (2015) and Perez-Rivas (2015) sequenced the entire coding sequence. As indicated by the reviewer, we modified the statement, eliminating the sentence regarding the intron-exon junctions and underlying the fact that in some previously published articles the forward primer used to determine the mutational status of USP8 encompasses the 664 residue (Cassarino 2018, Losa 2019) or is designed downstream of the 664 residue (Weigand 2019, Bujko 2019, Castellnou 2020), thus making not possible to identify the G664R variant.

Reference of Cassarino et al., 2018 has been added in the revised version of the manuscript as [37] and references numbers have been modified accordingly.

4- No study has detected changes in USP8 gene and protein expression in USP8 mutant adenomas {Weigand 2019, Ma 2014, Bujko 2019, Sesta 2020), thus there is so far no rationale to assume upregulation of USP8 in variant adenomas (see page 12, lines 484-485)

As pointed out by the reviewer no changes in USP8 gene and protein expression have been found in the majority of the studies on USP8-mutated tumors. Our assumption referred to results from our in vitro experiments that did not show any difference between mock transfected cells (endogenously expressing physiological levels of USP8) and USP8 wild type transfected cells (overexpressing USP8). We apologize if this statement was misleading and we have deleted the entire sentence from the revised text of the manuscript.

5- there are several grammar and conceptual errors (mouse adrenocortical tumor cells AtT-20, page 7, line 292)

“Mouse adrenocortical tumor cells AtT-20” cells have been corrected in “mouse corticotroph tumor cells AtT-20” (page 7, line 313). We carefully read the text checking for other grammar and conceptual errors and adjustments have been performed accordingly.

6- the paper could benefit from discussion of additional findings on USP8 pathophysiology in corticotrope adenomas, e.g., Martins 2020, Sesta 2020

Following the reviewer’s suggestion, we have revised the discussion of the manuscript by adding some comment regarding additional findings on USP8 pathophysiology in corticotrope adenomas found by Martins 2020 and Sesta 2020.

Page 12, lines 483-487: “It also has to be mentioned that in a Brasilian cohort no difference in the expression gene of markers involved in cell cycle regulation including p27/Kip1 has been found between wild-type and USP8 mutated groups [21] , suggesting that altered USP8 deubiquitinase activity might influence other pathways related to cell growth and proliferation.”

Page 11, lines 439-442: “Our clinical data are in line with those of Martins and colleagues showing that USP8 mutated tumors has a superior tumor size but similar clinical and biochemical phenotype than wild-type tumors [21].

Page 12, lines 487-494: “At the same time, gene expression profiles analysis in USP8 mutant and wild-type corti-cotrope tumors revealed that counter regulatory mechanisms may fine-tune the EGFR-MAPK pathway in tumor corticotroph cells [17]. Testing the hypothesis of corticotrope-specific USP8 substrates other than EGFR has provided remarkable insight into the biological behaviour of these cells. Indeed, the recent demonstration of the ACTH bio-synthetic pathway as a direct target of ubiquitination has raised the idea of targeting the ubiquitin proteasome system to modulate ACTH turnover both in USP8 mutant and wild-type tumors [17].”

7- given the small number of specimens carrying USP8 variants, statistics should be performed with non parametric tests and comparisons on 7 patients described with appropriate caution, especially as regards tumor size and recurrence, known to be linked per se.

As pointed out by the reviewer, a methodological description of the statistical analysis should have been clearly stated.

The M&M section of the manuscript has been revised as follow (page 5, lines 234-243):

“The results obtained with AtT-20 cells are expressed as the mean ± S.D. A paired two-tailed Student's t-test was used to detect the significance between two series of data. p<0.05 was accepted as statistically significant. Patients’ data are expressed as mean ± S.D. for normally distributed continuous variables, median and interquartile range (IQR) or range for non-Gaussian data and proportion for categorical parameters. The latter were analysed using the χ2 test or the Fisher exact test if the expected value was <5. Continuous parameters with normal distribution were compared using the t test and non-Gaussian data using the non-parametric test of Mann Whitney. Two-sided p-value was considered statistically significant when <0.05. All statistical analyses were performed using SPSS, version 26 (IBM, Chicago, IL).”

Moreover, since comparisons on 7 patients require appropriate caution, especially as regards tumor size and recurrence, known to be linked per se, the discussion has been revised as follow (page 11, lines 435-437):

“From a clinical point of view, more than half of USP8 mutation carriers had an ACTH secreting macroadenoma and were at higher risk of recurrence compared to WT patients, although our analysis could have been affected by the small number of patients.”

Reviewer 2 Report

The authors present an interesting manuscript investigating the somatic mutation profile of pituitary adenomas underlying Cushing’s disease. The authors investigated sequences at key hotspots in USP8, USP48 and BRAF genes in 60 patients (45 females). Using frozen and FFPE tissues, the authors first sequenced and confirmed canonical USP8 mutations in the 14-3-3 binding motif in 7 patients, USP48 mutations in 2 patients and none with BRAF mutations. The authors found a novel variant (c.2188G>A) that resulted in p.G644R that is upstream of the 14-3-3 binding motif.  This is a novel variant with a high predicted pathogenicity and has not been found in population databases. The authors then show increased ACTH secretion (95%) and cell proliferation with BrDU incorporation (21%) in ATT20 cells transiently transfected with G644R. The authors then showed increased pCREB and slight reduction in p27 with G644R transfection. The authors then report increased cleavage of G644R USP8 compared to wildtype protein much like increased cleavage reported with mutations in the 14-3-3 binding motif. They report that G644R mutant protein has decreased 14-3-3 interaction and does not localize to the nucleus.

This is an interesting manuscript that could be made stronger. The manuscript is well written and flows well. The authors address the shortcomings and weaknesses appropriately in the Discussion section.  Here are my detailed comments:

  1. What’s the allele frequency for G644R in the one sample tested?
  2. Anti-FLAG antibodies can detect full length and 40kD USP8 fragments. I recommend using anti-FLAG antibodies to complete the cleavage story.
  3. The authors need to convincingly show EGFR expression in their subclone of ATT20 cells. The entire pERK story hinges on EGFR expression in the cell type. Alternatively, the authors can co-transfect HeLa or COS7 cells with EGFR + G644R USP8.
  4. The lack of data from mock transfected cells is a huge weakness in the manuscript. Please include data from mock transfected cells.
  5. Authors need to show the EGFR and pEGFR before moving to downstream ERK signaling with EGF stim experiments.
  6. Again the cleavage story is incomplete. I strongly recommend adding entire gel blots, anti-FLAG and anti-Nterm USP8 antibody probes to convincingly demonstrate increased cleavage.
  7. Please clearly state the selection criteria and numbers measured for semi-quantitative results including PLA events.
  8. Please include patient derived adenoma samples stained for USP8 to demonstrate cytosolic localization of G644R USP8.

Author Response

Answer to reviewer

Reviewer #2

The authors present an interesting manuscript investigating the somatic mutation profile of pituitary adenomas underlying Cushing’s disease. The authors investigated sequences at key hotspots in USP8, USP48 and BRAF genes in 60 patients (45 females). Using frozen and FFPE tissues, the authors first sequenced and confirmed canonical USP8 mutations in the 14-3-3 binding motif in 7 patients, USP48 mutations in 2 patients and none with BRAF mutations. The authors found a novel variant (c.2188G>A) that resulted in p.G644R that is upstream of the 14-3-3 binding motif.  This is a novel variant with a high predicted pathogenicity and has not been found in population databases. The authors then show increased ACTH secretion (95%) and cell proliferation with BrDU incorporation (21%) in ATT20 cells transiently transfected with G644R. The authors then showed increased pCREB and slight reduction in p27 with G644R transfection. The authors then report increased cleavage of G644R USP8 compared to wildtype protein much like increased cleavage reported with mutations in the 14-3-3 binding motif. They report that G644R mutant protein has decreased 14-3-3 interaction and does not localize to the nucleus.

This is an interesting manuscript that could be made stronger. The manuscript is well written and flows well. The authors address the shortcomings and weaknesses appropriately in the Discussion section.  Here are my detailed comments:

  1. What’s the allele frequency for G644R in the one sample tested?

The G664R variant was identified in an ACTH-secreting pituitary tumor sample by Sanger sequencing. PCR amplification and direct sequencing were repeated three times. Since we could not perform NGS analysis on this sample due to the scarcity of tumor tissue, we could not calculate the variant allele frequency in the one sample tested by specific software for detect low-frequency variant. In order to show the presence of the variant in the sample, we added a supplementary figure (Suppl. Fig. 1) in the revised Supplementary data with the G664R variant in the electropherograms.

Suppl. Fig. 1. USP8 G664R variant identification by Sanger sequencing. PCR amplification and direct sequencing were repeated three times. Encoded amino acids are shown in one-letter code; mutant amino acid is represented in red.

Upon reviewer’s request one out of the three electropherograms image shown could be added to revised Fig.1 as Fig.1B (Fig. 1B would then become Fig.1C).

  1. Anti-FLAG antibodies can detect full length and 40kD USP8 fragments. I recommend using anti-FLAG antibodies to complete the cleavage story.

Wild type and mutants USP8 were cloned into pME-FLAG plasmid where the N-terminal of USP8 sequence was fused to the FLAG tag. Because of this, the anti-FLAG antibody can recognize the full length form of USP8 and the N-term 90 kDa USP8 fragment but not the C40 kDa active fragment. Similarly, our antibody against USP8 from Merck was designed against a synthetic peptide equivalent to the N-terminal of USP8, thus it can recognize the full length form of USP8 and the N-term 90 kDa USP8 fragment but not the C40 kDa active fragment. We apologize if we forget to include these information in the manuscript, we have now revised the Material and Methods section accordingly (page, 3, lines 131-133; page 4, lines 159, 163, 168-171).

Since both antibodies recognize the same bands of USP8 and the results obtained were superimposable we only showed Western blot images obtained with the anti-USP8 antibody. We apologize if  “FLAG-USP8” mistakenly appeared in figure 3A instead of “USP8 full length”. Figure 3A has been revised accordingly.

Moreover, given that the cleavage of USP8 results in the generation of N-tem 90 kDa and C40 kDa fragments in a ratio 1:1, we can assume that the levels of the N-term 90 kDa fragment would correspond to the levels of the C40 kDa fragment.

  1. The authors need to convincingly show EGFR expression in their subclone of ATT20 cells. The entire pERK story hinges on EGFR expression in the cell type. Alternatively, the authors can co-transfect HeLa or COS7 cells with EGFR + G644R USP8.

All experiments shown in this paper have been performed on an AtT-20 cells clone endogenously expressing EGFR, as demonstrated by preliminary RT-PCR and Western blot analysis shown in the images below.

Both primers used for RT-PCR were designed on exon junctions in order to avoid amplification of genomic DNA. RT-PCR reaction encompasses 5 exons. PCR products were Sanger sequenced to confirm they were murine EGFR.

Suppl. Fig. 2A. RT-PCR showing the presence of murine EGFR mRNA in AtT-20 cells. 1=AtT-20; 2=bone marrow (positive control); 3=100 bp marker; 4=NTC (no template control).

Suppl. Fig. 2B. Western blot analysis of EGFR protein expression. AtT-20 cells were tested for EGFR protein expression. 30 µg of total proteins extracted from AtT-20 cells at passages #4 and #10 were resolved by SDS page and Western blot analysis was performed with anti-EGFR antibody (Cell signalling, 1:1000 dilution).

Moreover, before starting all the experiments we tested the EGFR response to EGF stimulation in our AtT-20 cells by looking at ERK phosphorylation. As shown in the image below of a representative preliminary experiment, both 10 and 30 minutes of cells exposure to 50ng/ml EGF have led to substantial ERK phosphorylation increase  compared to unstimulated cells, suggesting the presence of an active EGFR system in our cell model.

Suppl. Fig. 2C. EGF-mediated increase of phospho-ERK1/2. AtT-20 cells were seeded in starved medium for24 h, then stimulated with EGF 50ng/ml for the indicated times and immediately chilled on ice. 30 µg of total proteins were resolved by SDS page and Western blot analysis was performed with antibodies anti phospho-ERK and anti-total ERK (Cell signalling, 1:1000 dilution).

These data have been added to the revised version of the supplementary materials section.

  1. The lack of data from mock transfected cells is a huge weakness in the manuscript. Please include data from mock transfected cells.

Upon reviewer’s suggestion, we revised figure 2a, b, c, corresponding figure legend (Page 9, lanes 361-363) and the results section by adding data from mock transfected cells. In particular:

  • page 8, lines 320-324: “We found that cells expressing G664R USP8 had increased ACTH secretion compared to empty vector (mock) control cells (+114.5±53.6, p<0.05) and those expressing WT USP8, and similar ACTH secretion compared to those expressing S718del USP8 and USP8-C40 (Figure 2A).;
  • page 8, lines 324: we deleted “(data not shown)” at the end of the sentence “No difference in ACTH secretion was observed in WT USP8 and empty vector (mock) transfected cells.”;
  • page 8, lines 326-329: “Our results showed that the transfection of USP8 G664R in AtT-20 cells promoted cell proliferation (+28.3±2.6%, p<0.001 vs mock), similarly to S718del USP8 (+42.8±22.4%, p<0.05 vs mock) and USP8-C40 (+24.6±16.2%, p<0.05 vs mock) (Figure 2B).”
  • page 8, lines 335-338: “In contrast, in cells expressing S718del and G664R USP8, a persistent activation of ERK was observed at 48 hours incubation with EGF (+71.5±10.5% S718del USP8 vs mock, p<0.05, +67.6±34.7%, G664R USP8 vs mock, p<0.05) (Figure 2C).

The abstract was modified accordingly (page 1, lines 40-42: “Transient transfection with the USP8 G664R variant resulted in a significant increase of ACTH release and cell proliferation (+114.5±53.6% and +28.3±2.6% vs empty vector transfected cells, p<0.05, respectively).”

  1. Authors need to show the EGFR and pEGFR before moving to downstream ERK signaling with EGF stim experiments.

We agree with the reviewer about the importance of showing data supporting EGFR expression in our AtT-20 cells subclone. To this regard, as reported in answer to comment number 3, preliminary RT-PCR and Western blot results demonstrating the expression and function of EGFR in AtT-20 cells used for our experiments have been added to the supplementary section of the revised manuscript (revised Suppl. Fig. 2A/B/C). 

  1. Again the cleavage story is incomplete. I strongly recommend adding entire gel blots, anti-FLAG and anti-Nterm USP8 antibody probes to convincingly demonstrate increased cleavage.

Here we show different images (acquired with different exposure time) of the same entire gel blot obtained with the anti USP8 antibody that we used to demonstrate the increased USP8 cleavage in USP8 G664R transfected cells (Fig.3A).

The low exposure time image below only allowed to detect the full length form of USP8

The high exposure time image below allowed to see the N-term 90 kDa USP8 fragment but the band corresponding to full length USP8 results oversaturated.

Thus, since it was not possible to clearly see both full length USP8 and N-term 90 kDa USP8 on the same image we selected a low exposure time image for USP8 full length and a high exposure time image for USP8 N-term (Fig.3A). However, upon reviewer’s request this entire gel blot may be added to the revised version of figure 3A.

Moreover, as reported in answer to question number 2 and in revised Materials and Methods, the antibody against FLAG can recognize the same USP8 forms (USP8 full length and USP8 N-term 90 kDa) as the antibody against N-terminal USP8. The same western blot results have been obtained with the two antibodies, thus we decided to omit the redundant anti-FLAG pictures.

  1. Please clearly state the selection criteria and numbers measured for semi-quantitative results including PLA events.

A detailed explanation of PLA analysis has been added to the materials and methods section of the revised manuscript as follow (page 5, lines 213-223):

“Proximity ligation events were quantified with NIH ImageJ software [27]. Briefly, cells were first searched using an excitation light of 405 nm (DAPI signal) and fine focus adjustment was performed switching the filter of the excitation light to FITC (488 nm). This procedure minimized photobleaching before image acquisition of PLA signals. Images of cells showing PLA spots were subjected to deconvolution protocols before quantification in order to exclude any possible unspecific signal from the analysis. Ten fields for each transfection condition were randomly chosen from three independent experiments. About 150 cells showing PLA signals were analysed for each condition for each experiment. About a total number of 1215±103 PLA spot were quantified for USP8 WT, with a mean of 7,2±0,5 PLA events/cell which was then considered as a reference for S718del USP8 and G664R USP8.”

  1. Please include patient derived adenoma samples stained for USP8 to demonstrate cytosolic localization of G644R USP8.

We agree with the reviewer about the importance of showing the G664R USP8 staining in the tumor tissue surgically removed from the patient, but unfortunately, due to the lack of remaining material, it was not possible to perform this immunohistochemical analysis. A comment in the discussion session of the revised manuscript has been added (page 13, lines 516-519): “Unfortunately, due to lack of material, we could not test the immunoreactivity of G664R USP8 in the tumor sample of the patient, thus the intracellular localization of G644R USP8 in human corticotropinomas remains to be fully demonstrated.”

Round 2

Reviewer 1 Report

As mentioned in my initial review, this paper does not read as study with an underlying hypothesis, rather a Case Report on a novel variant. The Authors perfomed some changes but obviously cannot address the underlying weakness.

The comments, both in the original and revised version, on clinical data in patients with  USP8-variant adenomas (7 patients in all) are a case in point.

Author Response

We thank the reviewer for the careful reading of our manuscript and for his/her comments and suggestions.

We draw on the editor's experience in evaluating the more suitable attribution for the present manuscript.

Reviewer 2 Report

This is a nice paper. The authors have adequately addressed all concerns. 

Author Response

We thank the reviewer for the careful reading of our manuscript and for her/his comments and suggestions. 

Round 3

Reviewer 1 Report

none